# Palmitic Acid and β-Hydroxybutyrate Induce Inflammatory Responses in Bovine Endometrial Cells by Activating Oxidative Stress-Mediated NF-κB Signaling

**DOI:** 10.3390/molecules24132421

**Published:** 2019-07-01

**Authors:** Peng Li, Lanzhi Li, Cai Zhang, Xi Cheng, Yi Zhang, Yang Guo, Miao Long, Shuhua Yang, Jianbin He

**Affiliations:** 1Key Laboratory of Zoonosis of Liaoning Province, College of Animal Science and Veterinary Medicine, Shenyang Agricultural University, Shenyang 110866, China; 2College of animal science, Henan University of Science and Technology, Luoyang 471003, China

**Keywords:** palmitic acid, β-hydroxybutyrate, inflammatory injury, NF-κB pathway, bovine endometrial cell

## Abstract

Ketosis is a nutritional metabolic disease in dairy cows, and researches indicated that ketonic cows always accompany reproductive problems. When ketosis occurs, the levels of non-esterified fatty acids (NEFAs) and β-hydroxybutyrate (BHBA) in the blood increase significantly. Palmitic acid (PA) is a main component of saturated fatty acids composing NEFA. The aim of this study was to investigate whether high levels of PA and BHBA induce inflammatory responses and regulatory mechanisms in bovine endometrial cells (BEND). Using an enzyme-linked immunosorbent assay, quantitative real-time PCR, and western blotting, we evaluated oxidative stress, pro-inflammatory factors, and the nuclear factor (NF)-κB pathway in cultured BEND cells treated with different concentrations of PA, BHBA, pyrrolidinedithiocarbamate (PDTC, an NF-κB pathway inhibitor), and N-acetylcysteine (NAC, an antioxidant). The content of malondialdehyde was significantly higher, the content of glutathione was lower, and antioxidant activity—glutathione peroxidase, superoxide dismutase, catalase, and total antioxidant capacity—was lower in treated cells compared with control cells. PA- and BHBA-induced oxidative stress activated the NF-κB signaling pathway and upregulated the release of pro-inflammatory factors. Moreover, PA- and BHBA-induced activation of NF-κB-mediated inflammatory responses was inhibited by PDTC and NAC. High concentrations of PA and BHBA induce inflammatory responses in BEND cells by activating oxidative stress-mediated NF-κB signaling.

## 1. Introduction

Ketosis is a nutritional metabolic disease that occurs in high-yielding dairy cows that exhibit a negative energy balance in the perinatal period, as characterized by higher concentrations of non-esterified fatty acids (NEFAs) and β-hydroxybutyrate (BHBA) in the blood. Ketosis is usually divided into clinical and subclinical ketosis according to the presence or absence of clinical manifestations [1]. When ketosis occurs, the levels of NEFAs and BHBA in the blood increase significantly, causing liver damage, oxidative stress, and a series of physiological metabolic disorders. These factors subsequently affect the immune function and reproductive performance of dairy cows, leading to other postpartum diseases [2]. Negative energy balance in the perinatal period is a major cause of this condition. Meanwhile, a link between clinical ketosis in dairy cows and decreased milk production, rumen stagnation, constipation, decreased feed intake, as well as loss of body condition, and an association between hyperketosis and increased levels of oxidative stress has also been revealed [3]. Ketogenic cows have a lower milk yield, lower milk fat content, and higher concentrations of BHBA, NEFAs, interleukin (IL)-6, tumor necrosis factor (TNF), and lactic acid in serum compared with healthy cows [4,5,6], and the antioxidant capacity of ketone-causing bovine bodies is also greatly reduced [7]. High levels of NEFAs and BHBA are also associated with inflammatory responses during diseases such as cow mastitis and metritis. Studies have also shown that cows with postpartum and subclinical endometritis in the perinatal period are significantly impaired in terms of blood neutrophil function and have higher concentrations of NEFAs and BHBA in the blood compared with uterine-healthy cows [8,9]. Endometritis also accompanied by changes in feed intake, and increased circulating concentrations of BHBA and NEFAs in the blood [10]. Oxidative damage to tissues and cellular components has also been found to be a primary or secondary cause of certain human diseases; however, the effect of oxidative stress on the health of animals, especially during periods of high metabolic activity, is much less understood [11].

The health of dairy cows during the perinatal period is very important, with health problems greatly reducing reproductive performance and promoting the development of various metabolic diseases, which in turn can lead to serious reproductive disorders. Ketosis is the most common metabolic disease during this period, having a large economic impact on the agriculture industry [12]. Understanding the pathogenic mechanisms that connect ketosis and uterine inflammation is therefore important.

Palmitic acid (PA), a saturated fatty acid produced via the conversion of excess carbohydrates, is commonly found in animals, plants, and microorganisms. PA is the most abundant saturated fatty acid among NEFAs, which are mainly composed of oleic acid (approximately 54%), PA (approximately 34%), and stearic acid (approximately 6%). Studies have shown that in hepatocytes [13] and muscle cells [14], elevated PA concentrations regulate apoptosis and autophagy through mitochondrial dysfunction and endoplasmic reticulum stress (mediated by increased oxidative stress) [15]. Meanwhile, PA was also found to upregulate expression of pro-inflammatory genes in macrophages [16], with endothelial cells undergoing chronic lipotoxicity under high concentrations of PA [17]. A previous study also revealed that saturated fatty acids (SFAs), but not unsaturated fatty acids (UFAs), induce nuclear factor (NF)-κB activation and expression of cyclooxygenase (COX)-2 and other inflammatory markers [18]. Overall, these studies suggest that PA and BHBA act as inflammatory stimulants to induce inflammatory responses in the endometrial cells of cows with ketosis. However, the specific mechanisms by which PA and BHBA induce genital tract inflammation are not fully understood.

NF-κB is a family of transcription factors that regulates various genes involved in important physiological processes such as survival, inflammation, and immune responses. In normal cells, NF-κB is present in the cytoplasm as an inactive heterodimer composed of two subunits, p50 and p65, forming a complex with the inhibitory protein IκBα. When cells are activated by certain inflammatory factors, IκBα is phosphorylated, causing rapid degradation followed by dissociation of NF-κB from IκBα. The phosphorylated (p)-IκBα is then rapidly degraded by the proteasome, causing translocation of NF-κB to the nucleus where it binds to specific DNA sequences present in target gene promoters [19]. It has also been suggested that activation of NF-κB is associated with inflammatory signaling and tumor development, such as upregulation of COX-2 and nitric oxide synthase; up-regulation of the inflammatory cytokines IL-6, IL-8, and TNF-a; and upregulation of the chemokines CCL2 and CXCL8 [20,21]. In addition, increased reactive oxygen species (ROS) in bovine hepatocytes is known to cause over-activation of NEFA-induced NF-κB inflammatory pathways, further increasing liver inflammatory damage in fatty-liver cows [22]. NF-κB acts as a major transcription factor, triggering inflammatory responses and regulating the expression of genes encoding various mediators [23]. In addition, cancer, autoimmune diseases, and inflammation can also lead to improper regulation of NF-κB [20,24]. However, whether the NF-κB pathway also regulates oxidative stress-induced uterine damage in cows remains unclear. This prompted us to explore the relationship between NF-κB-mediated signaling pathways and oxidative stress and inflammatory damage in bovine endometrial (BEND) cells.

The aim of this study was to investigate whether high levels of PA and BHBA activates the oxidative stress-mediated NF-κB signaling pathway, promoting the synthesis of inflammatory factors in BEND cells. The findings provide valuable insight into the pathogenic relationship between nutritional metabolic diseases and uterine inflammation.

## 2. Results

### 2.1. Effects of PA and BHBA on BEND Viability

As shown in Figure 1, at all time points, relative cell viability was decreased with increasing PA or BHBA concentration. However, the relative cell viability with 0.6 mmol/L BHBA was higher than in the control group (Figure 1B). A significant difference in cell viability was observed between different PA and BHBA concentrations after 24-h treatment (Figure 1A,B). Then, 24 h was selected as the treatment time for all subsequent experiments. At 24 h, based on the effect of different concentrations of PA and BHBA on BEND viability, 0.2, 0.4 (*P* < 0.05) and 0.6 mmol/L (*P* < 0.01) PA, and 0.6, 1.2 (*P* < 0.01) and 2.4 mmol/L (*P* < 0.01) BHBA were selected for subsequent assays of cellular oxidative status.

### 2.2. Oxidative Status

The content of glutathione (GSH) and the activity of catalase (CAT) and glutathione peroxidase (GSH-PX) decreased in the PA- and BHBA-treated groups compared with the control, reaching the lowest levels at 0.6 mM PA and 2.4 mM BHBA, respectively (Figure 2A,C,D). The activity of superoxide dismutase (SOD) in the PA- and BHBA-treated groups were not affected significantly compared with the control (Figure 2E). Levels of total antioxidant capacity (T-AOC) in the PA-treated groups were not decreased significantly compared with the control except in the 2.4 mM BHBA group (Figure 2F). Meanwhile, malondialdehyde (MDA) levels were increased compared with the control (Figure 2B). Therefore, 0.6 mM PA and 2.4 mM BHBA were used for all subsequent experiments.

### 2.3. RT-PCR Analysis

The gene expression data are shown in Figure 3. Levels of IL-6, IL-8, TNF-α, and NF-κB p65 mRNA increased significantly (*P* < 0.05 or *P* < 0.01) in both the PA (Figure 3A) and BHBA groups (Figure 3B) compared with the control. In addition, levels of IL-6, IL-8, TNF-α, and NF-κB p65 mRNA in the PA + NAC and PA + PDTC groups were decreased significantly (*P* < 0.05 or *P* < 0.01) compared with the PA group (Figure 3A). Meanwhile, the mRNA expression levels ofIL-6, IL-8, TNF-α, and NF-κB p65 in the BHBA + NAC and BHBA + PDTC groups were decreased significantly (*P* < 0.05 or *P* < 0.01) compared with the BHBA group (Figure 3B).

### 2.4. Levels of Pro-Inflammatory Factors in Cell Supernatants

Pro-inflammatory factor levels are presented in Figure 4. Although no significant differences in IL-6 and IL-8were observed between the PA group and control, or between the PA + NAC and PA + PDTC groups and PA group, levels were higher in the PA group than in the control, and were lower in the PA + NAC and PA + PDTC groups than in the PA group (Figure 4A,B). TNF-α levels were also significantly higher in the PA group (*P* < 0.05) than in the control and were significantly lower in the PA + NAC group (*P* < 0.05) than in the PA group (Figure 4C). Furthermore, significantly higher levels of IL-6, IL-8, and TNF-α were observed in the BHBA group than in the control (*P* < 0.01), although NAC and PDTC reduced this increase in a significant manner (*P* < 0.05 or *P* < 0.01) (Figure 4D–F).

### 2.5. NF-κB Signal Protein Expression

The protein expression data are shown in Figure 5. Protein levels of NF-κB p65 were increased significantly in the PA (*P* < 0.01) and BHBA group (*P* < 0.05) compared with the control (Figure 5A,D), but were decreased significantly in the PA + NAC (*P* < 0.01) and PA + PDTC group (*P* < 0.01) compared with the PA group (Figure 5A). They were also decreased significantly in the BHBA + NAC (*P* < 0.01) and BHBA + PDTC groups (*P* < 0.05) compared with the BHBA group (Figure 5D). Similarly, protein levels of p-IκBα were significantly higher in the PA (*P* < 0.01) and BHBA group (*P* < 0.05) than the control (Figure 5B,E), but were significantly lower in the PA + NAC (*P* < 0.01) and PA + PDTC groups (*P* < 0.05) than the PA group (Figure 5B). They were also significantly lower in the BHBA + NAC (*P* < 0.05) and BHBA + PDTC groups than the BHBA group (Figure 5E). Conversely, protein levels of IκBα decreased significantly in the PA (*P* < 0.05) and BHBA group (*P* < 0.05) compared with the control (Figure 5C,F), but increased significantly in the PA + NAC (*P* < 0.05) and PA + PDTC groups (*P* < 0.05) compared with the PA group (Figure 5C). They also increased significantly in the BHBA + NAC (*P* < 0.05) and BHBA + PDTC groups (*P* < 0.01) compared with the BHBA group (Figure 5F).

## 3. Discussion

Health problems in dairy cows during the perinatal period are of utmost importance because they can reduce reproductive performance and increase the risk of various metabolic diseases, resulting in significant reproductive disorders [25]. Among the common metabolic diseases that occur during this period, hyperketosis (which manifests as hyperketonemia and high free fatty acidemia) results in serious economic losses to the modern dairy industry [2]. Studies have shown that ketogenic cows also have higher concentrations of non-esterified fatty acids and β-hydroxybutyrate compared with nonlactating heifers [26]. Moreover, PA has been shown to induce apoptosis by increasing oxidative stress in nerve cells and astrocytes, in contrast to oleic acid and lauric acid [27]. As a long-chain 16:0 SFA, PA is the most common fatty acid in animals and plants, and it causes ROS overproduction in human neuroblastoma BE(2)-M17 cells [28]. In this study, we used high concentrations of PA and BHBA to induce oxidative stress and secretion of inflammatory factors and investigate the relationship between the release of inflammatory factors and the NF-κB signaling pathway. First, we used CCK-8 to determine the viability of the cells under different concentrations of PA and BHBA over a time gradient. The results revealed a significant effect of drug concentration at 24 h after treatment. This time point was therefore selected for all subsequent experiments, with PA concentrations of 0.2, 0.4, and 0.6 mmol/L and BHBA concentrations of 0.6, 1.2, and 2.4 mmol/L. This range was selected because each concentration caused an insignificant significant effect on cell viability after 24 h. In addition, we found that BHBA at 0.6 mmol/L has an enhancement effect on the vitality of BEND cells.

To verify whether PA and BHBA induced oxidative stress in BEND cells in vitro, we determined the content of MDA and GSH, and activity of CAT, GSH-PX, SOD, and T-AOC in the cell supernatant. Because MDA is a lipid peroxidation product, levels reflect the severity of cellular damage by oxygen free radicals [29]. Levels of SOD activity reflect a cell’s ability to scavenge oxygen free radicals, while GSH is the most important non-enzymatic antioxidant in the body and is required as a substrate for GSH-PX during the decomposition of hydroperoxide. Activity of GSH-PX and CAT in cells represents the ability to clear harmful H_2_O_2_. In addition, MDA and T-AOC can be used as an important basis to determine the balance between oxidants and antioxidants in animals [30]. It was previously suggested that PA stimulates lipid accumulation and oxidative stress in normal mouse hepatocytes, as well as stimulating an inflammatory response through the NOX4 and mitochondrial pathways [31]. Moreover, PA also stimulates oxidative stress responses in βTC6 cells [32], HepG2 cells [33] and HTR-8/SVneo cells [34]. Our study revealed that PA treatment promotes the production of MDA, while decreasing the content of GSH, and reducing the activity of CAT, GSH-PX, SOD, and T-AOC in BEND cells. BHBA treatment also resulted in an increase in MDA [3], decrease in GSH, and reduction in CAT, GSH-PX, SOD, and T-AOC activity. These results are consistent with a previous study on BHBA in cow primary hepatocytes [35]. Previous studies of our research group also showed that BHBA can cause oxidative stress [36], while this study further confirmed that BHBA can cause changes in cellular oxidation indexes. These findings confirm that PA and BHBA induce oxidative stress in BEND cells in vitro. This peroxidation state was also affected by the drug concentration; thus, 2.4 mmol/L BHBA and 0.6 mmol/L PA were selected for subsequent experiments because they had the largest effect on the peroxidation state. It was previously suggested that cows with blood BHBA > 1.2 mmol/L, glucose < 2.5 mmol/L, and NEFA > 0.5 mmol/L show obvious signs of ketosis, while those with blood BHBA < 0.60 mmol/L, glucose > 3.75 mmol/L, and NEFA < 0.4 mmol/L have no obvious signs [37]. This is also consistent with the in vitro ketosis model concentrations established in this study.

Oxidative stress-mediated inflammatory responses have long been recognized as an important cause of various inflammatory diseases in perinatal dairy cows [38]. Oxidative stress reportedly increases mRNA levels of inflammatory factors such as TNF-α, IL-6, and IL-1β [39,40]. In our study, both PA and BHBA induced oxidative stress, increasing mRNA levels of inflammatory factors IL-8, IL-6, and TNF-α. We therefore evaluated levels of IL-8, IL-6, and TNF-α proteins by ELISA, revealing similar trends, which suggested that PA and BHBA promote inflammation, consistent with previous studies [41,42]. NAC, a commonly used antioxidant, exerts anti-oxidative activity and protects cells by scavenging oxygen free radicals under oxidative stress damage [43]. NAC has also been shown to inhibit NF-κB activation [44]. In the present study, RT-PCR and ELISA showed significant inhibition of the PA- and BHBA-induced increase in inflammatory factors IL-8, IL-6, and TNF-α after NAC addition. Similarly, the PA- and BHBA-induced increase in IL-8, IL-6, and TNF-α was also significantly inhibited after the addition of PDTC. Related studies have suggested that PDTC inhibits the activation of NF-κB by specifically suppressing release of the inhibitory subunit IκB from the latent cytoplasmic form of NF-κB [45,46]. The findings of the present study suggest that the inflammatory response induced by oxidative stress involves activation of signaling pathways that are dependent on the release of inflammatory factors.

The NF-κB transcription factor family is the most important regulator of transcriptional immune and inflammatory responses [47]. NF-κB also plays an important role in processes such as development, cell growth and survival, and proliferation, as well as numerous pathological conditions. It was also revealed that excessive ROS can activate the NF-κB pathway. Conversely, NF-κB was found to regulate the transcription of related genes, thereby affecting ROS levels [48]. PA activates the NF-κB transcription factor and induces IL-6 and TNF-α expression in 3T3-L1 adipocytes [49], while BHBA was found to promote the expression of pro-inflammatory factors in calf hepatocytes by activating the NF-κB signaling pathway [35]. To determine whether the release of pro-inflammatory factors IL-8, IL-6, and TNF-α is also induced by the NF-κB signaling pathway, we examined the protein expression of IκBα, p-IκBα, and NF-κB p65, and the mRNA levels of NF-κB p65. The results showed that when high concentrations of PA and BHBA were applied to BEND cells, protein expression levels of p-IκBα and NF-κB p65 increased significantly, whereas after the addition of the antioxidant NAC and pathway inhibitor PDTC, the PA- and BHBA-induced increase in p-IκBα and NF-κB p65 protein expression was significantly inhibited. Unlike the expression of p-IκBα, the expression of IκBα showed the opposite pattern. Moreover, the mRNA levels of NF-κB p65 were consistent with the protein results. These results suggest that PA- and BHBA-induced oxidative stress activates the NF-κB signaling pathway in BEND cells, which in turn induces a pro-inflammatory response. However, the specific mechanism of oxidative stress-induced activation of the NF-κB signaling pathway in BEND cells requires further study.

## 4. Materials and Methods

### 4.1. Chemical Reagents and Antibodies

Dulbecco’s modified Eagle’s medium (DMEM)/F12 medium, trypsin (with EDTA), antibiotics (50 U/mL penicillin; 50 μg/mL streptomycin), fetal bovine serum (FBS), and phosphate-buffered saline (PBS) were purchased from HyClone Laboratories Inc. (Logan, UT, USA). PA, β-hydroxybutyrate, N-acetylcysteine (NAC, an antioxidant), and pyrrolidine dithiocarbamate (PDTC, an NF-κB inhibitor) were purchased from Sigma-Aldrich (Saint Louis, MO, USA). p-IκBα, IκBα, NF-κB p65, and β-actin antibodies were purchased from Cell Signaling Technology (Danvers, MA, USA), and histone H3 antibody was purchased from Abcam (Cambridge, MA, USA). Cell culture flasks were purchased from Thermo Fisher Scientific (Waltham, MA, USA), and six-well plates, 96-well plates, and filters were purchased from Nalge Nunc (Rochester, NY, USA).

### 4.2. Cell Culture

An immortalized cell lines of BEND cells were purchased from the BeNa Culture Collection (BNCC340413, Beijing, China). Cells were cultured in DME/F-12 medium (HyClone, USA) supplemented with 10% fetal bovine serum (BioInd, Beit HaEmek, Israel), 2% penicillin/streptomycin (HyClone, USA) at 37 °C in an incubator with 95% air and 5% CO_2_ atmosphere. The culture medium was replaced daily, and subculture was conducted by trypsinization with 0.05% trypsin until the cells reached 85–90% confluence. The cells were cultured in cell flasks and prepared for the following experiments.

### 4.3. Cell Viability Assay

A Cell Counting Kit-8 (CCK-8, Beijing Solarbio Technology Co., Ltd., Beijing, China) was used to determine the viability of BEND cells treated with different concentrations of PA or BHBA for different lengths of time. Briefly, a suitable amount of cell suspension (2 × 10^6^ cells/mL) was inoculated onto a 96-well plate (100 μL/well) and pre-cultured in a 37 °C, 5% CO_2_ incubator. After the cells were completely attached, 0.1, 0.2, 0.3, 0.4, 0.6, and 0.8 mmol/L PA or 0.6, 1.2, 2.4, 4.8, 7.2, and 9.6 mmol/L BHBA was added and allowed to incubate for 6, 12, 24 or 48 h. The experiments for each treatment concentration of PA and BHBA were replicated 3 times. Finally, 10 µL of CCK-8 solution was added to each well, and the plates were placed in an incubator for 3 h. Absorbance at 450 nm was measured using a microplate reader (Bio-Rad Instruments, Hercules, CA, USA).

### 4.4. Oxidative Stress Parameters Analysis

BEND cells (2 × 10^6^ cells/mL) were seeded in six-well plates (2 mL/well). Confluent (80–90%) cells were treated with different concentrations of PA or BHBA for 24 h. Each treatment was replicated 3 times. The antioxidant status of the BEND cells was determined based on glutathione (GSH) and malondialdehyde (MDA) contents, enzyme activities of glutathione peroxidase (GSH-PX), superoxide dismutase (SOD), and catalase (CAT), and the total antioxidant capacity (T-AOC) of the cell supernatant. Standard commercial kits were used (Nanjing Jiancheng Bioengineering Institute, Nanjing, China) and the experiments were carried out according the manufacturer’s instructions.

### 4.5. RNA Extraction and Quantitative Real-Time PCR

Total RNA from BEND cells was extracted using a Total RNA Extraction Kit (TaKaRa Biotechnology Co., Ltd., Tokyo, Japan) according to the manufacturer’s instructions. The RNA concentration and purity were determined using a nucleic acid protein detector (Bio-Rad Instruments). Reverse transcription of the extracted RNA into single-stranded cDNA was carried out using a reverse transcription kit (TaKaRa Biotechnology Co., Ltd.) according to the manufacturer’s instructions. Quantitative real-time PCR (RT-PCR) was performed using SYBR Premix Ex Taq II (TaKaRa Biotechnology Co., Ltd.) to assess mRNA expression levels of TNF-α, IL-6, IL-8, and NF-κB p65. All amplifications were repeated three times. RT-PCR data were calculated using the gene expression formula 2^−ΔΔCt^. The relative expression of each gene was normalized to β-actin levels. The primers used for RT-PCR are listed in Table 1. Primers for IL-8 and TNF-α were designed using Primer Express Version 5.0 software (New York, NY, USA), and primers for NF-κB p65, IL-6, and β-actin were identical to those used by Shi et al. [40]. All primers were commercially synthesized by Sangon Biotech Institute Co., Ltd. (Beijing, China).

### 4.6. Enzyme-Linked Immunosorbent Assay

BEND cells (2 × 10^6^ cells/mL) were seeded in six-well plates (2 mL/well). Confluent (80–90%) cells were treated with different drugs for 24 h. Each treatment was replicated 3 times. The cell supernatants were centrifuged at 3000 rpm for 10 min to extract the liquid phase for assays of the pro-inflammatory cytokines TNF-α, IL-6, and IL-8 using enzyme-linked immunosorbent assay (ELISA) kits (Beijing Dongge Weiye Technology Co., Ltd., Beijing, China) according to the manufacturer’s instructions.

### 4.7. Western Blot Analysis

After discarding the cell culture medium, the cells were washed with pre-cooled PBS before being collected. The cell culture flasks were then placed on ice, and 2 mL of pre-cooled PBS was added before scraping the cells from the culture flask using a cell scraper. Total cellular proteins and nuclear proteins were extracted using a protein extraction kit (contain phosphatase inhibitors) and nuclear protein extraction kit (Beijing Solarbio Technology Co., Ltd.), respectively, according to the manufacturer’s instructions. The concentration of protein extracted was quantified using a BCA kit (Beijing Solarbio Technology Co., Ltd.).

Proteins (50 µg of protein per lane) were separated by 10% sodium dodecyl sulfate-polyacrylamide gel electrophoresis (Shanghai EpiZyme Biological Technology Co., Ltd., Shanghai, China) according to the standard protocol and electrophoretically transferred onto polyvinylidene fluoride membranes (0.45 μm). Membranes were soaked in 5% skim milk (Beijing Solarbio Technology Co., Ltd.) and blocked for 1 h in a constant-temperature shaker at room temperature. The membranes were incubated with NF-κB p65 (1:4000 *v/v*), IκBα (1:3000 *v/v*), p-IκBα (1:1000 *v/v*), β-actin (1:3000 *v/v*), or histone H3 (1:1000 *v/v*) primary antibodies diluted in Tris-HCl buffer solution containing Tween 20 (TBST) overnight at 4 °C. The next day, the membranes were washed five times with TBST for 10 min each at room temperature with shaking, incubated with appropriate peroxidase-conjugated secondary antibodies diluted in 5% skim milk for 90 min at room temperature with shaking, and then washed five times for 10 min each. Finally, the membranes were exposed using enhanced chemiluminescence solution (New Cell & Molecular Biotech Co., Ltd., Suzhou, China) and MicroChemi (DNR Bio Imaging Systems, Neve Yamin, Israel) was used to visualize the relative band intensities.

### 4.8. Statistical Analysis

All values are expressed as the mean ± standard error of the mean (SEM). Statistical analyses were carried out using SPSS 17.0 software (SPSS Inc., Chicago, IL, USA). Significant differences were evaluated by analysis of variance (ANOVA), and Tukey’s post hoc test was evaluated for significance difference. Statistical significance was defined as *P* < 0.05 or *P* < 0.01.

## 5. Conclusions

The findings of this study suggest that high concentrations of PA and BHBA induce oxidative stress-mediated activation of the NF-κB signaling pathway in BEND cells, thereby regulating the expression and release of inflammatory factors IL-8, IL-6, and TNF-α, leading to inflammatory damage.

## Figures and Tables

**Figure 1 molecules-24-02421-f001:**
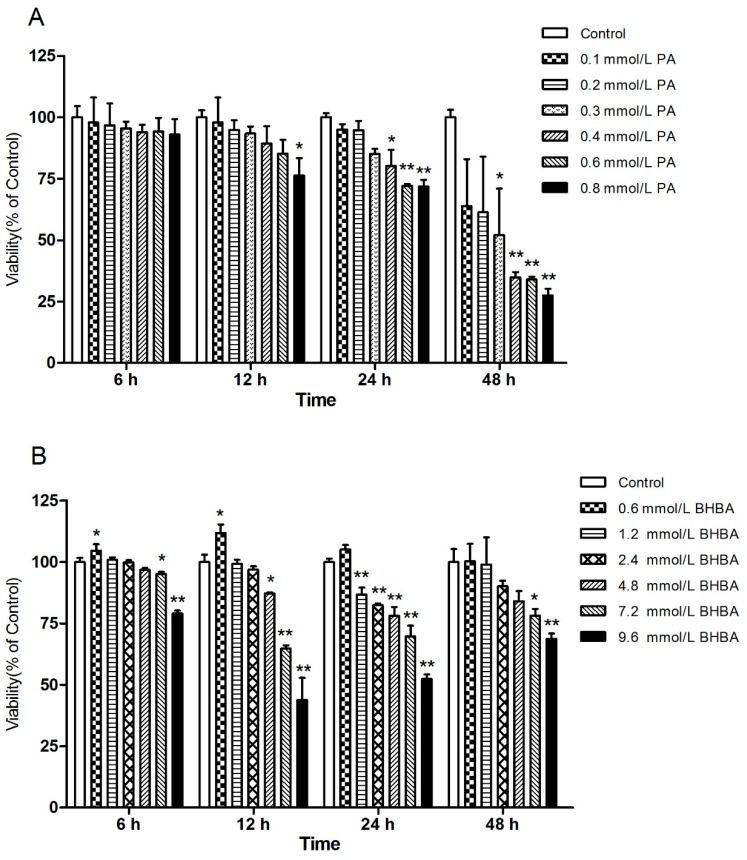
Effects of palmitic acid (PA) and β-hydroxybutyrate (BHBA) on cell viability. (**A**) Bovine endometrial cells (BEND) cells were treated with PA (0.1, 0.2, 0.3, 0.4, 0.6, or 0.8 mmol/L) for 6, 12, 24, or 48 h. (**B**) BEND cells were treated with BHBA (0.6, 1.2, 2.4, 4.8, 7.2, or 9.6 mmol/L) for 6, 12, 24, or 48 h. * *P* < 0.05, ** *P* < 0.01 vs. the control.

**Figure 2 molecules-24-02421-f002:**
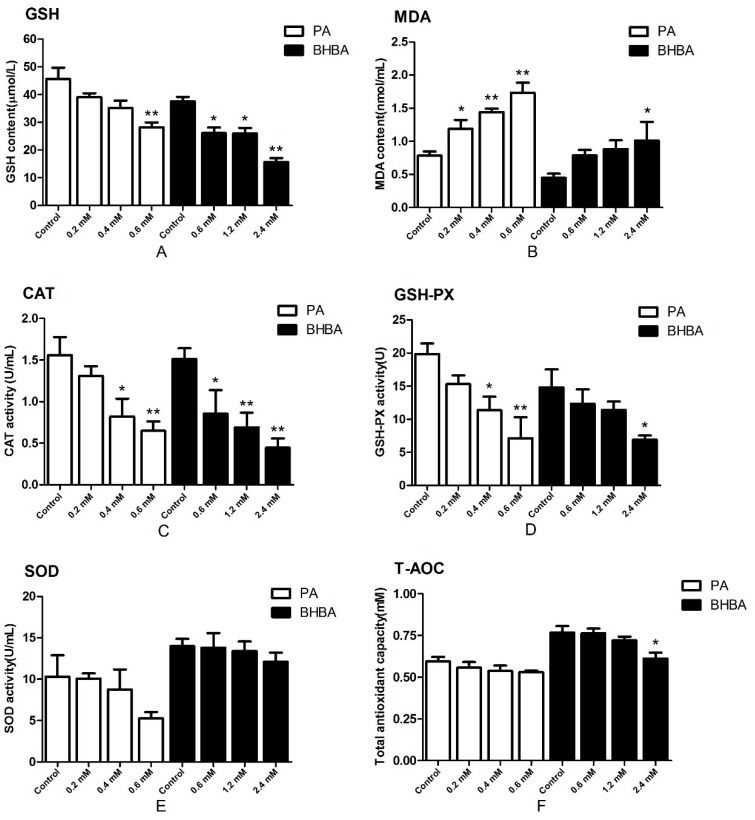
Effects of PA and BHBA on oxidative status. Contents of (**A**) glutathione (GSH) and (**B**) malondialdehyde (MDA), and activities of (**C**) catalase (CAT), (**D**) glutathione peroxidase (GSH-PX), (**E**) superoxide dismutase (SOD), and (**F**) total antioxidant capacity (T-AOC) in BEND cells treated with PA (0.2, 0.4, or 0.6 mM) or BHBA (0.6, 1.2, or 2.4 mM) for 24 h. Data are expressed as the mean ± SEM. * *P* < 0.05, ** *P* < 0.01 vs. the control.

**Figure 3 molecules-24-02421-f003:**
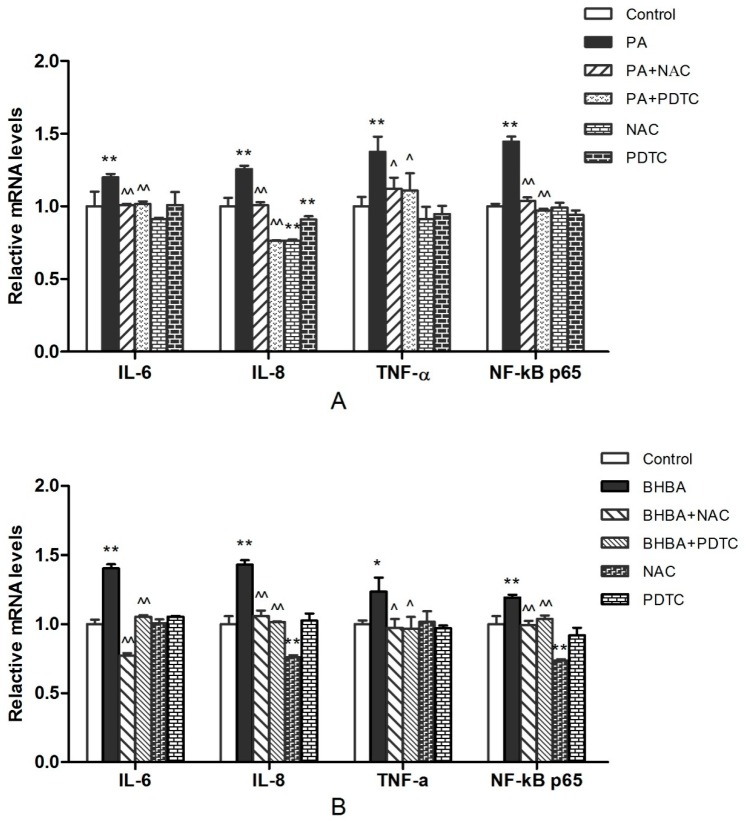
RT-PCR analysis of related gene expression. (**A**) BEND cells were treated with 0.6 mM PA, 0.6 mM PA + 10 mM NAC (PA + NAC), 0.6 mM PA + 10μM PDTC (PA + PDTC), 10 mM NAC, or 10 μM PDTC for 24 h. (**B**) BEND cells were treated with 2.4 mM BHBA, 2.4 mM BHBA + 10 mM NAC (BHBA + NAC), 2.4 mM BHBA + 10 μM PDTC (BHBA + PDTC), 10 mM NAC, or 10 μM PDTC for 24 h. Data are expressed as the mean ± SEM. * *P* < 0.05, ** *P* < 0.01 vs. the control group; ^ *P* < 0.05, ^^ *P* < 0.01 vs. the PA or BHBA group.

**Figure 4 molecules-24-02421-f004:**
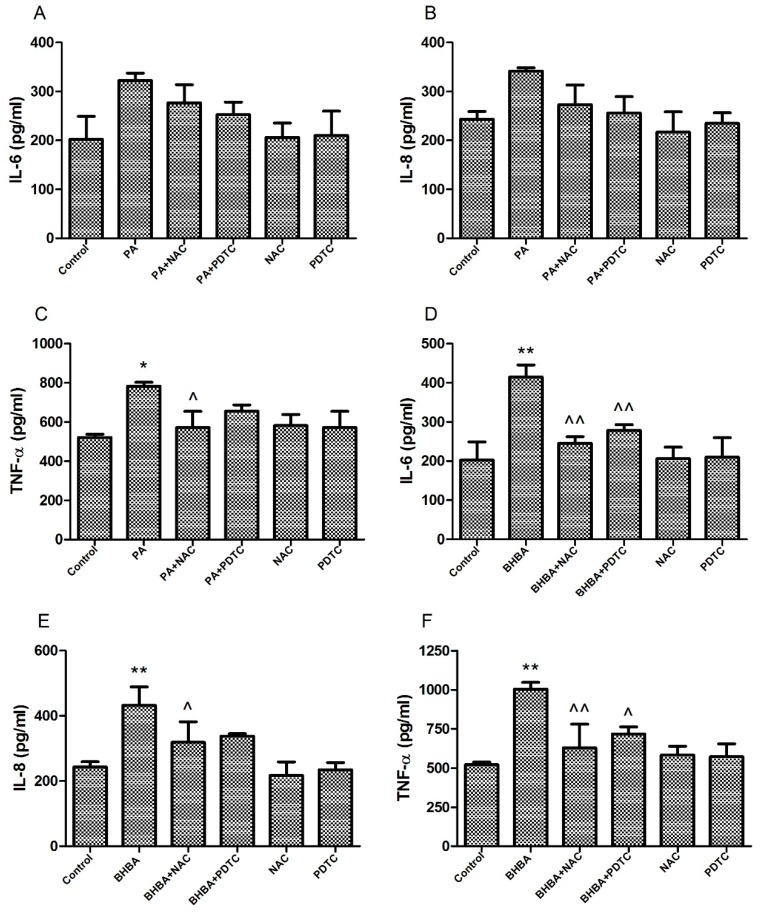
Levels of pro-inflammatory factors in the cell supernatants. (**A**–**C**) BEND cells were treated with 0.6 mM PA, 0.6 mM PA + 10 mM NAC (PA + NAC), 0.6 mM PA + 10 μM PDTC (PA + PDTC), 10 mM NAC, or 10 μM PDTC for 24 h. (**D**–**F**) BEND cells were treated with 2.4 mM BHBA, 2.4 mM BHBA + 10 mM NAC (BHBA + NAC), 2.4 mM BHBA + 10 μM PDTC (BHBA + PDTC), 10 mM NAC, or 10 μM PDTC for 24 h. Data are expressed as the mean ± SEM. * *P* < 0.05, ** *P* < 0.01 vs. the control; ^ *P* < 0.05, ^^ *P* < 0.01 vs. the PA or BHBA group.

**Figure 5 molecules-24-02421-f005:**
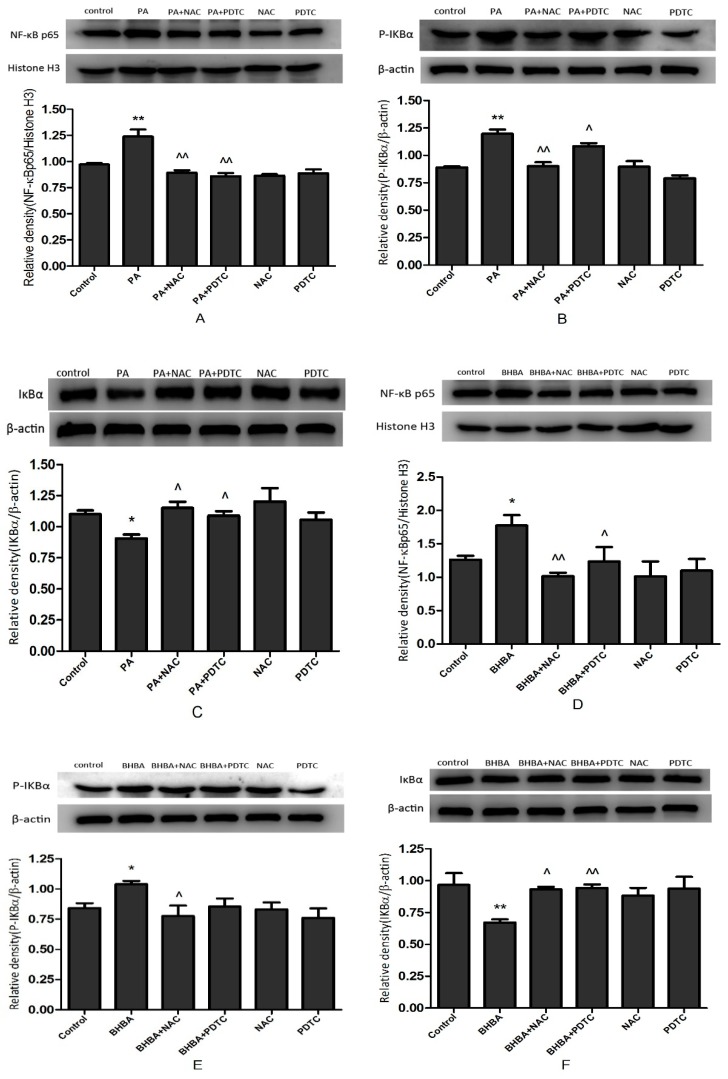
NF-κB signal protein expression. (**A**–**C**) BEND cells were treated with 0.6 mM PA, 0.6 mM PA + 10 mM NAC (PA + NAC), 0.6 mM PA + 10 μM PDTC (PA + PDTC), 10 mM NAC, or 10 μM PDTC. (**D**–**F**) BEND cells were treated with 2.4 mM BHBA, 2.4 mM BHBA + 10 mM NAC (BHBA + NAC), 2.4 mM BHBA + 10 μM PDTC (BHBA + PDTC), 10 mM NAC, or 10 μM PDTC. (**A**,**D**) Relative protein expression levels of NF-κB p65 normalized to histone H3. (**B**,**E**) Relative protein expression levels of p-IκBα normalized to β-actin. (**C**,**F**) Relative protein expression levels of IκBα normalized to β-actin. Data are expressed as the mean ± SEM. * *P* < 0.05, ** *P* < 0.01 vs. the control; ^ *P* < 0.05, ^^ *P* < 0.01 vs. the PA or BHBA group.

**Table 1 molecules-24-02421-t001:** Primers for RT-PCR.

Gene	Primer Sequence (5′→3′)	Length of Amplified Fragment (bp)
IL-8	For: TTAGGCAGACCTCGTTTCCAT Rev: ATGACTTCCAAGCTGGCTGTT	235
TNF-α	For: GGTCAACATCCTGTCTGCCA Rev: ACTGAGGCGATCTCCCTTCT	130
NF-κB p65	For: AGGACCAACCAGACCG Rev: TGTCACCAGGCGAGTTAT	240
IL-6	For: AACGAGTGGGTAAAGAACGC Rev: CTGACCAGAGGAGGGAATGC	144
β-actin	For: GCCCTGAGGCTCTCTTCCA Rev: GCGGATGTCGACGTCACA	101

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
