# Peer review of "Palmitic Acid and β-Hydroxybutyrate Induce Inflammatory Responses in Bovine Endometrial Cells by Activating Oxidative Stress-Mediated NF-κB Signaling"

_molecules, 2019, doi:10.3390/molecules24132421_

Round 1
Reviewer 1 Report
The authors examined the effects of Palmitic acid (PA) and β-hydroxybutyrate (BHBA) induced inflammatory responses in bovine endometrial cells. They explained that Palmitic acid (PA) and β-hydroxybutyrate (BHBA)-induced oxidative stress through activating NF-κB signaling pathway, oxidative stress markers, pro-inflammatory factors and decreasing antioxidant enzymes. Despite this study conveys interesting piece of information, major revision should carry.
1. The significance of using bovine endometrial cells in this study, particularly to examine the oxidative and antioxidant parameters?
2. The data presented for the activity of SOD and the levels of T-AOC did not consistent with the content of the manuscript. The respective figures should modify in the revise paper.
3. What is the reason behind to the results of NAC or PTDC alone did not potential when compared to their effect with stimulators?
4. The term of NF-κB p65 and its results should consistently use throughout the paper. Moreover, the results shown in Fig. 5A and D have to check again carefully.
5. This paper requires complete English edition by a native English speaker.
Author Response
Dear Reviewer:
On behalf of my co-authors, we thank you very much for your valuable time for this article, we appreciate editor and reviewers very much for your positive and constructive comments and suggestions on our manuscript entitled “Palmitic Acid and β-hydroxybutyrate Induce Inflammatory Responses in Bovine Endometrial Cells by Activating Oxidative Stress-mediated NF-κB Signaling”. Below is our response to comments resulting in a number of clarifications.
1. The significance of using bovine endometrial cells in this study, particularly to examine the oxidative and antioxidant parameters?
Response: Ketosis is a nutritional metabolic disease in dairy cows, and researches indicated that ketonic cows always accompany with reproductive problems. When ketosis occurs, the levels of non-esterified fatty acids (NEFAs) and β-hydroxybutyrate (BHBA) in the blood increase significantly. PA is a main component of saturated fatty acids composing non-esterified fatty acids. Above all, we focus on the effect of PA and BHBA on BEND, in order to confirm the pathogenic relationship between nutritional metabolic diseases (ketosis) and uterine inflammation.
2. The data presented for the activity of SOD and the levels of T-AOC did not consistent with the content of the manuscript. The respective figures should modify in the revise paper.
Response:We revised this section in the revision.
3. What is the reason behind to the results of NAC or PTDC alone did not potential when compared to their effect with stimulators?
Response:In the group treated with NAC or PDTC alone, the cells were in a normal physiological state due to the absence of PA or BHBA, and thus had no effect on cell indexes. But when PA and BHBA added in the media with NAC or PDTC simultaneously, oxidative stress and inflammatory response may induced by PA and BHBA and then their effect emerged. Forthermore, treated with NAC or PDTC alone also used as a positive control, and the results after acting alone on the cells were consistent with previous reports:
Pan, X.; Wu, X.; Yan, D.; Peng, C.; Rao, C.; Yan, H., Acrylamide-induced oxidative stress and inflammatory response are alleviated by N-acetylcysteine in PC12 cells: Involvement of the crosstalk between Nrf2 and NF-kappaB pathways regulated by MAPKs. Toxicology letters 2018, 288, 55-64.
Bolego, C.; Lu, C.-W.; Lin, Y.; Lei, Y.-P.; Wang, L.; He, Z.-M.; Xiong, Y., Pyrrolidine dithiocarbamate ameliorates endothelial dysfunction in thoracic aorta of diabetic rats by preserving vascular DDAH activity. Plos One 2017, 12, (7), e0179908.
4. The term of NF-κB p65 and its results should consistently use throughout the paper. Moreover, the results shown in Fig. 5A and D have to check again carefully.
Response: We revised this section in the revision and redescribed the results of Fig. 5A and D.
5. This paper requires complete English edition by a native English speaker.
Response: This article has been edited by a native English speaker.
We look forward to hearing from you regarding our submission. We would be glad to respond to any further questions and comments that you may have.
Thank you and best regards.
Yours sincerely,
Lanzhi Li
Reviewer 2 Report
Li et al. investigate the mechanisms of Palmitic acid (PA) and β-hydroxybutyrate (BHBA) induced inflammatory responses in bovine endometrial cells (BEND). They evaluated oxidative stress, pro-inflammatory factors, and the nuclear factor (NF)-κB pathway in cultured BEND cells treated with different concentrations of PA, BHBA, pyrrolidine dithiocarbamate (PDTC, an NF-κB pathway inhibitor), and N-acetylcysteine (NAC, an antioxidant). The authors found that PA- and BHBA-induced oxidative stress via inducing NF-κB signaling pathway and pro-inflammatory factors. Though this study carries some important aspect in inflammatory science, still there several issues need to be clarified.
1. At first, what is rationality of this study and why bovine endometrial cells (BEND) have been used to this study, especially to study about oxidative and antioxidant parameters?
2. It is mentioned that the activity of SOD decreased in the PA- and BHBA-treated groups compared with the control, reaching the lowest levels at 0.6 mM PA and 2.4 mM BHBA, respectively, but in the figure the activity of SOD is not significantly decreased. This needs clarification and the figure should replace with the text.
3. Similarly, the levels of T-AOC were also not decreased as the it is stated “lower in the PA- and BHBA-treated groups compared with the control”. This need to clarify.
4. The sentence “mRNA increased significantly or highly significantly (P < 0.05 or P < 0.01)” has to modify meaningfully.
5. The sentence “In addition, a significant or highly significant decrease in IL-6, IL-8, TNF-α, and NF-κB p65 mRNA expression levels (P < 0.05 or P < 0.01) was observed in the PA+NAC and PA+PDTC groups compared with the PA group (Fig. 3A), and in the BHBA+NAC and BHBA+PDTC groups compared with the BHBA group (Fig. 3B)” need thorough edition.
6. Almost no results show the pretreatment of NAC or PTDC alone is effective and it is a big contradiction. How the inhibitors alone without inducers have no effect, instead they are effective when combined with inducers.
7. Did PA not induce cytokines TNF-alpha and IL-8 significantly?
8. Whether it is total protein of NF-κB p65 or phosphorylation of NF-κB p65 significantly higher in the PA and BHBA group compared with the control (Fig. 5A and D)?
9. The section “4.4. Oxidation state analysis” should be replaced by Oxidative stress parameters analysis.
10. It is inconvincible to use NF-κB inhibitor PTDC and NAC to test the expression of cytokines. These PTDC and NAC should have been used to test on the oxidative stress markers, like MDA, GHS, CAT, SOD and GPx.
11. This article written in very poor English, and certainly needs to be edited by a native English speaker before being consider for publication.
Author Response
Dear Reviewer:
We thank you very much for your valuable time for this article. We appreciate the very useful comments on our manuscript entitled “Palmitic Acid and β-hydroxybutyrate Induce Inflammatory Responses in Bovine Endometrial Cells by Activating Oxidative Stress-mediated NF-κB Signaling” from the reviewers. We agree with these suggestions and have revised the manuscript accordingly. Below is our response to comments resulting in a number of clarifications.
1. At first, what is rationality of this study and why bovine endometrial cells (BEND) have been used to this study, especially to study about oxidative and antioxidant parameters?
Response: Ketosis is a nutritional metabolic disease in dairy cows, and researches indicated that ketonic cows always accompany with reproductive problems. When ketosis occurs, the levels of non-esterified fatty acids (NEFAs) and β-hydroxybutyrate (BHBA) in the blood increase significantly. PA is a main component of saturated fatty acids composing non-esterified fatty acids. Above all, we focus on the effect of PA and BHBA on BEND, in order to confirm the pathogenic relationship between nutritional metabolic diseases (ketosis) and uterine inflammation.
2. It is mentioned that the activity of SOD decreased in the PA- and BHBA-treated groups compared with the control, reaching the lowest levels at 0.6 mM PA and 2.4 mM BHBA, respectively, but in the figure the activity of SOD is not significantly decreased. This needs clarification and the figure should replace with the text.
Response:We revised this section in the revision.
3. Similarly, the levels of T-AOC were also not decreased as the it is stated “lower in the PA- and BHBA-treated groups compared with the control”. This need to clarify.
Response:We revised this section in the revision.
4. The sentence “mRNA increased significantly or highly significantly (P < 0.05 or P < 0.01)” has to modify meaningfully.
Response: We revised this section in the revision.
5. The sentence “In addition, a significant or highly significant decrease in IL-6, IL-8, TNF-α, and NF-κB p65 mRNA expression levels (P < 0.05 or P < 0.01) was observed in the PA+NAC and PA+PDTC groups compared with the PA group (Fig. 3A), and in the BHBA+NAC and BHBA+PDTC groups compared with the BHBA group (Fig. 3B)” need thorough edition.
Response: We revised this section in the revision.
6. Almost no results show the pretreatment of NAC or PTDC alone is effective and it is a big contradiction. How the inhibitors alone without inducers have no effect, instead they are effective when combined with inducers.
Response:In the group treated with NAC or PDTC alone, the cells were in a normal physiological state due to the absence of PA or BHBA, and thus had no effect on cell indexes. But when PA and BHBA added in the media with NAC or PDTC simultaneously, oxidative stress and inflammatory response may induced by PA and BHBA and then their effect emerged. Forthermore, treated with NAC or PDTC alone also used as a positive control, and the results after acting alone on the cells were consistent with previous reports:
Pan, X.; Wu, X.; Yan, D.; Peng, C.; Rao, C.; Yan, H., Acrylamide-induced oxidative stress and inflammatory response are alleviated by N-acetylcysteine in PC12 cells: Involvement of the crosstalk between Nrf2 and NF-kappaB pathways regulated by MAPKs. Toxicology letters 2018, 288, 55-64.
Bolego, C.; Lu, C.-W.; Lin, Y.; Lei, Y.-P.; Wang, L.; He, Z.-M.; Xiong, Y., Pyrrolidine dithiocarbamate ameliorates endothelial dysfunction in thoracic aorta of diabetic rats by preserving vascular DDAH activity. Plos One 2017, 12, (7), e0179908.
7. Did PA not induce cytokines TNF-alpha and IL-8 significantly?
Response: In the RT-PCR analysis, our results showed that PA can significantly induce the mRNA expression of cytokines TNF-alpha (P<0.01) and IL-8 (P<0.01). Similarly, in the ELISA assay, our results showed that PA can significantly promote the content of cytokine TNF-alpha (P<0.05) (Fig. 4C). Although the content of cytokine IL-8 in the PA group was increased, the difference was not significant compared with the control group (P>0.05) (Fig. 4B).
8. Whether it is total protein of NF-κB p65 or phosphorylation of NF-κB p65 significantly higher in the PA and BHBA group compared with the control (Fig. 5A and D)?
Response:We measured the total NF-κB p65 protein content in nuclear proteins. It is total protein of NF-κB p65 significantly higher in the PA and BHBA group compared with the control (Fig. 5A and D).
9. The section “4.4. Oxidation state analysis” should be replaced by Oxidative stress parameters analysis.
Response:We revised this section in the revision.
10. It is inconvincible to use NF-κB inhibitor PTDC and NAC to test the expression of cytokines. These PTDC and NAC should have been used to test on the oxidative stress markers, like MDA, GHS, CAT, SOD and GPx.
Response: In our previous experiments, we have demonstrated that PA and BHBA can induce oxidative stress in BEND cells. In subsequent experiments, we mainly wanted to understand whether the oxidative stress induced by PA and BHBA had an effect on the contents of cytokines TNF-α, IL-6, IL-8, and whether it was related to the NF-κB signaling pathway.
11. This article written in very poor English, and certainly needs to be edited by a native English speaker before being consider for publication.
Response: This article has been edited by a native English speaker.
We look forward to hearing from you regarding our submission. We would be glad to respond to any further questions and comments that you may have.
Thank you and best regards.
Yours sincerely,
Lanzhi Li
Round 2
Reviewer 1 Report
The authors have responded well. The paper can be accepted in present form.
Reviewer 2 Report
Most of my queries are clarified. The paper can be accepted now.